# Documenting contributions to scholarly articles using CRediT and *tenzing*

**Alex O. Holcombe**[1]*, **Marton Kovacs**[2], **Frederik Aust**[3,4], **Balazs Aczel**[2]

**1** School of Psychology, University of Sydney, Sydney, Australia, **2** Institute of Psychology, ELTE, Eotvos Lorand University, Budapest, Hungary, **3** University of Cologne, Cologne, Germany, **4** University of Amsterdam, Amsterdam, Netherlands

* alex.holcombe@sydney.edu.au

## Abstract

Scholars traditionally receive career credit for a paper based on where in the author list they appear, but position in an author list often carries little information about what the contribution of each researcher was. "Contributorship" refers to a movement to formally document the nature of each researcher's contribution to a project. We discuss the emerging CRediT standard for documenting contributions and describe a web-based app and R package called *tenzing* that is designed to facilitate its use. *tenzing* can make it easier for researchers on a project to plan and record their planned contributions and to document those contributions in a journal article.

## Introduction

Scholarly journal articles evolved from letters penned by individuals reporting scientific observations or experiment results. These letters listed only a single author, and it was clear that that person was claiming credit for all aspects of the work reported.

Today, over three hundred years later, most science is done by groups of people, not by lone individuals [1]. Different members of the team usually have different roles. Yet until recently, journals still operated as if there was no need to provide any information other than a list of names—the author list. Some information could be tentatively inferred from the order of names in the list, but how order is determined reflects often-unwritten practices around authorship that can be obscure to people outside a subfield and can differ substantially between labs [2].

When uncertain, people fall back on their prior beliefs. This is unfortunate for junior authors who do not have many papers to their name: when people see a list of authors with no explicit indication of who did what, they may give an outsize amount of credit to the senior author.

Fortunately, over the last few decades, many journals have begun to encourage, and some to require, that teams give some indication of who did what in the work reported by a paper. In some journals, this is done in a brief "Author Note" or "Author Information" section [e.g., 3]. Thanks to this development, researchers are more likely to get the specific recognition they deserve.

**Data Availability Statement:** The source code is archived at http://doi.org/10.5281/zenodo.3993411.

**Funding:** The authors received no specific funding for this work.

**Competing interests:** The authors have declared that no competing interests exist.

The included information would ideally be utilized by funders of scientists to allocate resources more effectively, so that teams with the right combination of skills would more often be supported. Moreover, those who hire scientists, such as universities and research institutes, should be able to assemble more effective teams for particular disciplines and projects.

Unfortunately, these potential benefits have been held back by a lack of standardization. Without a consistent vocabulary for describing what each researcher did in a project, and without a structured format for that information, it is difficult to aggregate across papers the type of contributions a researcher makes. For institutions and funders interested in supporting the right combinations of people, it is difficult to tally the sorts of contributions typically involved in different sorts of projects.

This issue is also faced by business and industry, where some solutions were devised. For commercial music for example, the recording industry uses an International Standard Musical Work Code (ISWC). This contains metadata for musical works that provide the identities of contributors and indicates whether they served the roles of, for example, composer, lyricist, or arranger [4,5]. A search of the associated ISWC database allows people to find the works that a musician has contributed to and what their role was in each work (http://iswcnet.cisac.org/).

In scientific research, roles may not be as clear cut as typical in the music industry. Nonetheless, useful distinctions can be made, such as contributions to the analysis of data versus to the drafting of a manuscript, or to the acquisition of data.

## CRediT

In 2014, the first formal taxonomy was developed for scientific research—CRediT, the Contributor Role Taxonomy [6]. CRediT defines fourteen different types of contributions (Table 1), and over the last several years, it has been taken up by hundreds of journals [7] and dozens of publishers (see http://credit.niso.org/adopters/) and been endorsed by a number of journal editors [8].

The use of CRediT not only can provide better documentation of the contributions of individual researchers, but also it enables meta-scientific research, such as into the different distribution of contributions indicated for women and men [9].

To facilitate researcher reporting of contributorship information in manuscripts and journal articles, we created *tenzing*, a web app and R package [10] for researchers and publishers. In the following, we will review how journals are currently using and reporting CRediT information. We then explain how *tenzing* can facilitate researcher and journal use of CRediT. Finally, we describe broader issues associated with CRediT contributorship that should be addressed as fields move forward with the usage of contributorship.

## How publishers are using CRediT

The CRediT standard includes a specification for how to report contributorship information in the metadata that is associated with manuscript webpages (JATS-XML). But many publishers do not yet have the capability to do this. For example, it appears that none of the organizations behind preprint servers currently create CRediT metadata in JATS-XML format. In such cases, it can be useful for researchers to publish CRediT information in plain text in their manuscripts. Many journals make no mention of CRediT but ask researchers to indicate what each author did in the "Author Note" or similar section of the manuscript. Researchers can use CRediT to do this, in their preprints and in their submitted manuscripts.

An increasing number of scientific journals offer authors forms to indicate which CRediT category each author contributed to. For example, in the submission interface of *eLife*, authors encounter an array of checkboxes to indicate which category each author contributed to (Fig 1).

**Table 1. Contributor roles according to the Contributor Role Taxonomy (CRediT) [6], information available online at http://credit.niso.org/.**

| Contributor role | Description |
|---|---|
| Conceptualization | Ideas; formulation or evolution of overarching research goals and aims |
| Data curation | Management activities to annotate (produce metadata), scrub data and maintain research data (including software code, where it is necessary for interpreting the data itself) for initial use and later re-use. |
| Formal analysis | Application of statistical, mathematical, computational, or other formal techniques to analyze or synthesize study data. |
| Funding acquisition | Acquisition of the financial support for the project leading to this publication. |
| Investigation | Conducting a research and investigation process, specifically performing the experiments, or data/evidence collection. |
| Methodology | Development or design of methodology; creation of models. |
| Project administration | Management and coordination responsibility for the research activity planning and execution. |
| Resources | Provision of study materials, reagents, materials, patients, laboratory samples, animals, instrumentation, computing resources, or other analysis tools. |
| Software | Programming, software development; designing computer programs; implementation of the computer code and supporting algorithms; testing of existing code components. |
| Supervision | Oversight and leadership responsibility for the research activity planning and execution, including mentorship external to the core team. |
| Validation | Verification, whether as a part of the activity or separate, of the overall replication/reproducibility of results/experiments and other research outputs. |
| Visualization | Preparation, creation and/or presentation of the published work, specifically visualization/data presentation |
| Writing–original draft | Preparation, creation and/or presentation of the published work, specifically writing the initial draft (including substantive translation) |
| Writing–review & editing | Preparation, creation and/or presentation of the published work by those from the original research group, specifically critical review, commentary or revision–including pre- or post-publication stages |

PLOS journals provide a similar facility (Fig 2), as do over 1200 Elsevier journals (https://www.elsevier.com/about/press-releases/corporate/elsevier-expands-credit-approach-to-authorship).

Many authors encounter the CRediT roles for the first time when they are submitting to a journal. Or even if an author has used CRediT for a previous paper, they may be unlikely to explicitly consider these roles for a new paper until the time of journal submission. From multiple perspectives, not considering contributor roles until the time of submission is not ideal.

By the time an author submits a manuscript, the associated research project sometimes was completed months or even years before. At the time of journal submission, memory of each

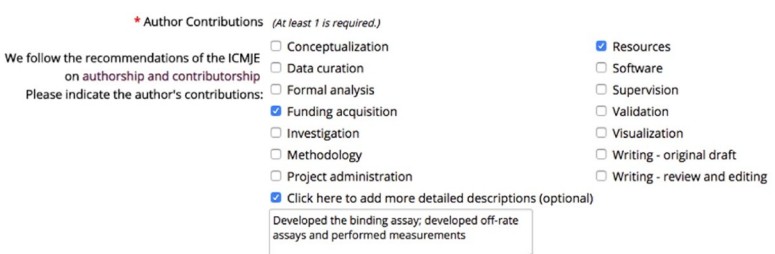

**Fig 1. The journal *eLife*'s interface for indicating contributions when submitting a manuscript, available online at https://elifesciences.org/inside-elife/f39cfcf5/enabling-the-contributor-roles-taxonomy-for-author-contributions.**

collaborator's contributions may be fuzzy. Ideally, authors will arrive at a consensus regarding who did what. But even if memories and records are adequate for this task, establishing such a consensus necessitates interruption of the submission of the manuscript until the submitting author hears from all the other authors and works to resolve any disagreement about various contributions, such as who contributed to the original draft of the manuscript.

Unfortunately, there is reason to believe that, when not discussed until after project completion, the rate of disagreement regarding author contributions may be high. Surveys suggest that between between a third and two-thirds of researchers have been involved in authorship disagreements [11–14]. In many fields, the submitting author is often the most junior author. This is typically the case when a PhD student submits her first paper, for example. Yet a student or other junior author is not in the best position to arbitrate disputes or push back on project contributors who may be overclaiming regarding their contribution [15]. For this and other reasons, there are many recommendations that authors communicate more about authorship expectations and roles, and that they should do so at the beginning of a project [16–19]. This may be even more important when the manuscript is to provide not only a list of author names, but also a specification of each author's contributions.

Most authorship disputes are settled informally, but still may leave some people bitter at being excluded, or resentful that some people were included on an authorship list without any evidence they deserved it. The same likely applies to disputes over which contribution

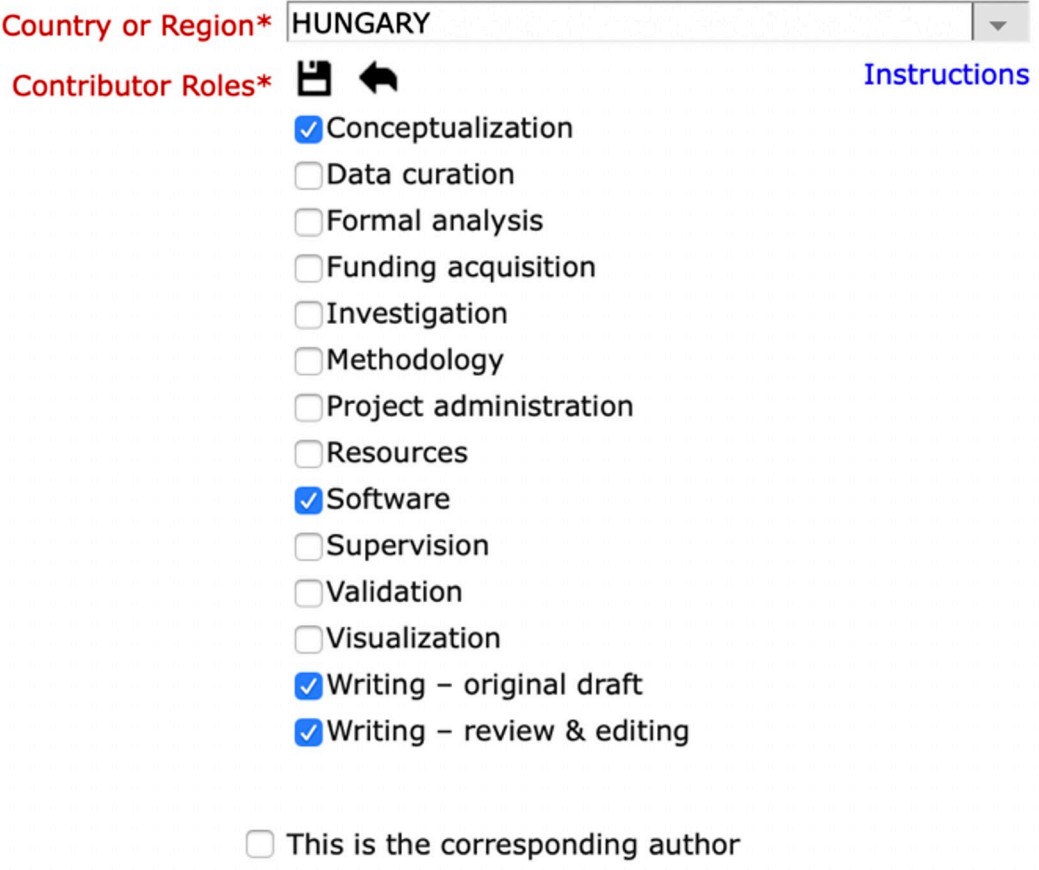

**Fig 2. The PLOS journals' interface for indicating contributions when submitting a manuscript.** It appears when one is asked to enter information about each author.

categories a researcher contributed to. It is probably best to get some agreement on these at the beginning of a project, so that researchers can proceed with some confidence around both what they are expected to do and what kind of credit they will get for it.

To facilitate project and credit attribution planning, an "authorship grid" system was described by Philippi et al. [20]. Each row of the grid is a task category or high-level responsibility associated with the project, and the columns are the researchers. At the intersection of the rows and columns, researchers indicate the more specific tasks they plan to perform, if any, in that category. This approach is likely very useful for complex projects. For CRediT-using journals, this needs to be translated into CRediT information, which *tenzing* can facilitate.

## How *tenzing* helps authors use CRediT

*tenzing* is a web app and associated R package that allows researchers to record contributorship information at any time, for eventual provision to a journal. The app is named after the mountaineer Tenzing Norgay, who together with Edmund Hillary was the first to reach the summit of Mount Everest. Norgay arguably received less credit than was appropriate given his contribution.

Here we will describe the use of *tenzing* solely in terms of the web app (https://marton balazskovacs.shinyapps.io/tenzing/), although one can also use it via the underlying R package (https://github.com/marton-balazs-kovacs/tenzing)—full documentation for *tenzing* can be found online at https://marton-balazs-kovacs.github.io/tenzing/.

Use of *tenzing* starts with a spreadsheet template (provided as a Google Sheet, http://bit.ly/tenzingTemplate, but one can also use it in any spreadsheet editor, such as Excel). For a given research project, researchers make a copy of the template and then, in the rows, enter the names of their collaborators (Fig 3). One column is dedicated to each of the fourteen CRediT categories, to be checked off to indicate which categories each researcher contributed to. Because some CRediT categories are not entirely self-explanatory, one can hover the cursor over the column names to see some additional defining information.

Around the time of the start of a project, a lead researcher may choose to send the link to the Sheet to all those involved, who can then indicate the areas they plan to contribute to. At the end of the project, or when plans change during the project, this Sheet can be revisited. Google Sheet services track the changes made in the template, thus by visiting the version history one can review the evolution of contributorship roles throughout the project.

When the researchers are ready to submit to a journal, they upload their filled-out spreadsheet to the *tenzing* app. They can then click a button to generate any of various outputs.

For CRediT, *tenzing* outputs a brief report in the form of a list indicating which contributor did what (Fig 4). This can be pasted into the section known at some journals as the Author Note. It is particularly appropriate for journals whose publishing platform does not support the machine-readable CRediT metadata. For example, the journal *Collabra*: *Psychology* encourages researchers to provide CRediT information in the "Author Contributions" section, because their publisher has not yet implemented creation of CRediT metadata in the article contents.

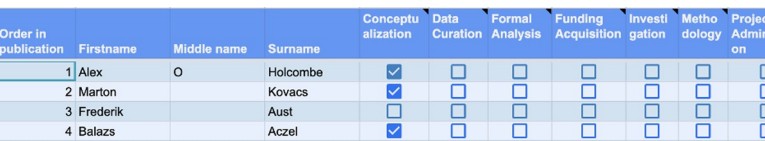

**Fig 3. Partial screenshot of the spreadsheet template (http://bit.ly/tenzingTemplate).**

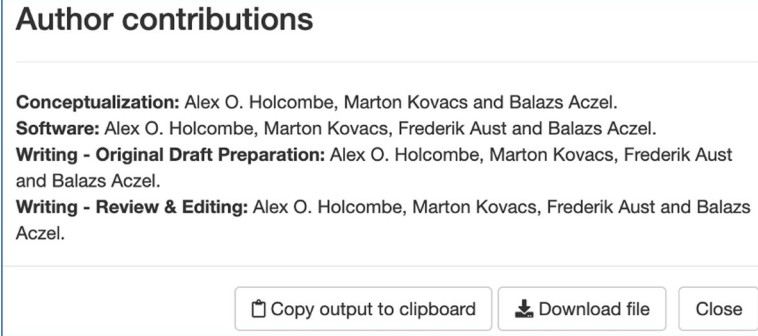

**Fig 4. Screenshot of the *tenzing* window that provides a report of author contributions.**

The publishing platforms used by dozens of publishers can include CRediT metadata in JATS-XML-format in the journal article webpages (see http://credit.niso.org/implementing-credit/). *tenzing* can generate this JATS-XML information itself for users to download (Fig 5). Ideally, researchers would be able to upload this to a journal submission portal when submitting their manuscript, obviating the need to fill in arrays of checkboxes for each contributor. Unfortunately, at present no journal is capable of processing the uploaded JATS-XML, although a few publishers have privately indicated that they're interested in adding support for this.

Some researchers write manuscripts in R Markdown and use the *papaja* package [21] to generate manuscripts in APA format for submission to a journal. *tenzing* generates author metadata in YAML-format, which can be included in the R Markdown file. *papaja* then includes the CRediT information in the Author Note section of the APA-formatted manuscript.

The current user interface for *tenzing* is shown in Fig 6, although its design is likely to evolve–a usability study is presently underway.

An additional output provided by *tenzing* is unrelated to CRediT: a list of the authors' names, with annotations indicating the institutions they are affiliated with, formatted to be suitable to paste into the title page of a manuscript file (Fig 7). For manuscripts with large numbers of authors, this can substantially reduce the time required to create the title page.

The current version of *tenzing* has various limitations, such as only allowing entry of one affiliation per author. Addressing this and a few other features is currently planned, with

The Journal Article Tag Suite (JATS) is an XML format used to describe scientific literature published online. Find out more about JATS XML

```xml
<?xml version="1.0" encoding="UTF-8"?>
<contrib-group>
  <contrib>
    <name surname="Aczel" given-names="Balazs"/>
    <role vocab="credit" vocab-identifier="http://dictionary.casrai.org
    <role vocab="credit" vocab-identifier="http://dictionary.casrai.org
    <role vocab="credit" vocab-identifier="http://dictionary.casrai.org
    <role vocab="credit" vocab-identifier="http://dictionary.casrai.org
  </contrib>
  <contrib>
    <name surname="Aust" given-names="Frederik"/>
    <role vocab="credit" vocab-identifier="http://dictionary.casrai.org
```

**Fig 5. A screenshot of a portion of the JATS-XML output provided by *tenzing*.**

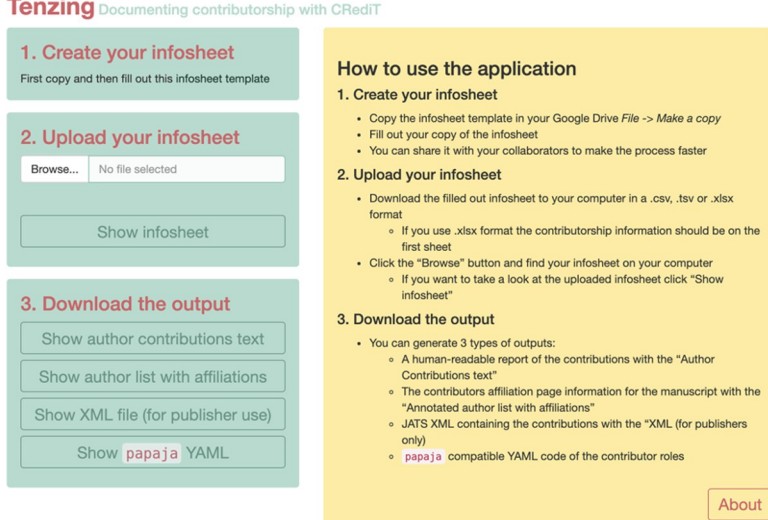

**Fig 6. A screenshot of the *tenzing* app.** The bottom portion of both sides describes the four outputs that *tenzing* provides.

updates regarding progress available at the development site (https://github.com/marton-balazs-kovacs/tenzing/issues). User interface professionals have provided some suggestions, which will likely result in improvements to the app's design and usability. *tenzing* is open source [10], and researchers and other community members are invited to contribute to *tenzing* development by posting feature requests and bug reports at the Github issues page (https://github.com/marton-balazs-kovacs/tenzing/issues) or by contacting the corresponding author.

## The future of CRediT

The CRediT standard was primarily designed to allow researchers to indicate what type of contribution they made. However, it also has a facility that allows one to indicate the *degree* of contribution. Specifically, one can optionally indicate whether each contributor to a particular category played a "lead", "equal", or "supporting" role in the associated work. It appears that most journals that use CRediT have opted not to use this feature, at least not yet. Editorial Manager, a journal platform used by thousands of journals, has integrated the degree of contribution feature but as a specific configuration, and most journals using Editorial Manager currently do not appear to have activated it.

An unresolved issue with CRediT's degree of contribution facet is how it should be used. It seems likely that if the "equal" degree is used, it must be used for multiple co-authors as it may not make sense when applied to just one. This is not currently addressed, however, by the

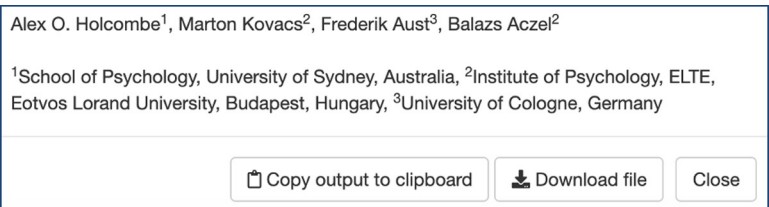

**Fig 7. A screenshot of the author list and affiliation output screen.**

CRediT documentation, nor are other possible constraints such as whether "equal" can be used as an intermediate indicator in cases where there are already authors with the "lead" and "supporting" labels. In addition, there is no indication to publishers of how they should indicate degrees of contribution in the machine-readable JATS-XML associated with journal articles, although Aries Systems, the creator of Editorial Manager, has done this by using the "specific-use" attribute (Caroline Webber, personal communication, 8 July 2020).

The degree of contribution under-specification is one of the issues that will likely be addressed by the group convened by the American National Information Standards Organization to formalize CRediT as an ANSI/NISO standard (https://niso.org/press-releases/2020/04/niso-launches-work-contributor-role-taxonomy-credit-initiative). For now, we have chosen to not yet implement the degree of contribution feature in *tenzing*.

## The future of contributorship

The number of contributors to the average scientific paper has steadily increased over the last several decades [22,23]. In part, this has occurred because as knowledge in an area increases, specialization facilitates further advances. Some forms of research today, such as systematic reviews and meta-analyses, are based on bringing together large amounts of evidence from the literature. Library professionals contribute to some such projects with sophisticated searches of papers and databases. For other projects, technicians provide invaluable guidance regarding equipment, programmers create needed software, statisticians provide statistical advice, and informaticists create visualizations or collate information from databases. With science increasingly depending on these tasks getting done, funders need to be able to assess what sorts of projects have most benefited from specialists in order to resource science most effectively. However, people in these specialist roles are often not included in author lists, making it difficult to determine the number of specialists contributing to various projects.

One obstacle to greater inclusion of specialist contributors is the current state of journal authorship guidelines. The authorship guidelines for thousands of journals are based on the International Committee of Medical Journal Editors. These guidelines stipulate that only those who contribute to the writing or revising of a manuscript are eligible for authorship [24]. Journals should consider expanding authorship eligibility, for example by adopting the proposal of McNutt et al. [8] to eliminate the writing requirement and endorse the use of CRediT [25].

Some fields, such as genomics, already have a tradition of including groups, often known as consortia, on an author list, without enumeration of individual researcher names. This is often used to indicate those who only contributed data, which is a useful alternative to making that particular distinction with CRediT [26].

CRediT is not a good fit for all disciplines or even all projects within a discipline [27]. An ontology of roles that is both broader than those of CRediT and also more specific has been developed by the National Center for Data to Health, an initiative of the National Center for Advancing Translational Sciences (NCATS) at the National Institutes of Health [28]. The scheme is called the Contributor Role Ontology (CRO, https://data2health.github.io/contributor-role-ontology/), and it extends the CRediT ontology to include more than fifty roles, including "specimen collection", "librarian", "community engagement", "coordination", and "software testing" [28–30]. Given the adoption of CRediT that has already occurred, we anticipate that improvements will occur via extensions or generalizations such as CRO. The CRO scheme could be integrated into *tenzing* in the future.

If author contributions to a journal's articles are explicitly indicated by a contributorship taxonomy such as CRediT or CRO, how should one think about the order of authorship? One might expect order to still be used for communicating the relative amount that different

authors contributed, despite its limitations due to ambiguity around interpreting the meaning of first author and last author in different fields and cultures. However, note that CRediT also allows an indication of degree of contribution, beyond just how many categories a researcher contributed to. Specifically, where multiple individuals serve in the same role, the degree of contribution can optionally be specified as 'lead', 'equal', or 'supporting', but as described in the previous section, the proper usage of as well as the metadata for this has not yet been fully specified in the CRediT standard.

Deciding on order of authorship may get more and more difficult as the number of authors increases. Having a discussion among the researchers to decide this, without a clear decision process, may be unwieldy. Some have suggested a points system for different types of contributions. The American Psychological Association online authorship resources site for several years has included an example "scorecard" that assigns different types of contributions different numbers of points [31]. For CRediT, one such points system has been created by Mojtaba Soltanlou [32]. However, the relative value of different sorts of contributions likely differs across projects.

A critically important document for communicating contributions to scholarship is the CV. Traditionally, the extent of different authors' contributions is communicated entirely by the order of authorship. In the future, however, we anticipate that funders or individual researchers will move to CVs that communicate the nature of the contributions made to each journal article. The Rescognito site [33] has created experimental visualizations, as did Ebersole, Adie, & Cook in a SIPS hackathon [25] with a bar graph indicating, for each CRediT category, how many papers a researcher contributed to.

Another piece of infrastructure already supporting CRediT usage is the ORCiD database and metadata for identifying researchers and linking them to their papers and other scholarly contributions [34]. Usage has grown rapidly, with over 7,000 papers a month indexed in Crossref because at least one author used ORCiD [35]. The ORCiD registry includes CRediT information. While *tenzing* could potentially pull author information such as name, email and affiliation from the ORCiD database rather than requiring manual entry, the selection of the information to import can have complications that require user intervention (for example, one might need to include an old affiliation and not the current one). A prototype shiny app available at https://colomb.shinyapps.io/contributorlist_creator/ facilitates that [36] and is now compatible with *tenzing*, as it can be used to create an infosheet one can further complete manually before uploading it into *tenzing*.

With adoption of CRediT growing rapidly, it is becoming more urgent to attend to any problems being encountered in its use or with the standard itself. The NISO effort to formalize CRediT will include a solicitation of feedback, which will be an important opportunity for the scholarly community to shape how contributorship information is recorded. We hope that the usage of CRediT facilitated by *tenzing* during the feedback period will result in a greater understanding of what about CRediT should be prioritized for refinement or change.

## Acknowledgments

We thank the Society for the Improvement of Psychological Science (SIPS) and the participants in the 2019 SIPS Hackathon on contributorship [25] for discussion.

## Author Contributions

**Conceptualization:** Alex O. Holcombe, Marton Kovacs, Balazs Aczel.

**Software:** Alex O. Holcombe, Marton Kovacs, Frederik Aust, Balazs Aczel.

**Writing – original draft:** Alex O. Holcombe, Marton Kovacs, Frederik Aust, Balazs Aczel.

**Writing – review & editing:** Alex O. Holcombe, Marton Kovacs, Frederik Aust, Balazs Aczel.

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
