## [Decision Letter · Decision Letter 0]

7 Sep 2020

PONE-D-20-23271

Documenting contributions to scholarly articles using CRediT and tenzing

PLOS ONE

Dear Dr. Holcombe,

Thank you for submitting your manuscript to PLOS ONE. After careful consideration, we feel that it has merit but does not fully meet PLOS ONE’s publication criteria as it currently stands. Therefore, we invite you to submit a revised version of the manuscript that addresses the points raised during the review process.

The manuscript deserves further discussion on specific advantages tenzing has over traditional methods in its aim of finding solutions to the problem of authors' contributions to scientific articles.

We look forward to receiving your revised manuscript.

Kind regards,

Bright Nwaru

Academic Editor

PLOS ONE

Additional Editor Comments:

None

Journal Requirements:

Reviewers' comments:

Reviewer's Responses to Questions

**Comments to the Author**

1. Is the manuscript technically sound, and do the data support the conclusions?

Reviewer #1: Yes

Reviewer #2: Yes

Reviewer #3: Yes

Reviewer #4: Yes

Reviewer #5: No

2. Has the statistical analysis been performed appropriately and rigorously? 

Reviewer #1: N/A

Reviewer #2: N/A

Reviewer #3: N/A

Reviewer #4: N/A

Reviewer #5: N/A

3. Have the authors made all data underlying the findings in their manuscript fully available?

Reviewer #1: Yes

Reviewer #2: Yes

Reviewer #3: Yes

Reviewer #4: Yes

Reviewer #5: Yes

4. Is the manuscript presented in an intelligible fashion and written in standard English?

Reviewer #1: Yes

Reviewer #2: No

Reviewer #3: Yes

Reviewer #4: No

Reviewer #5: Yes

5. Review Comments to the Author

Reviewer #1: The article summarizes advantages and limitations of traditional methods (e.g. “Author Note”, “Author information”) which have been used by scientific journals for many years. Then the authors briefly describe Contributor Role Taxonomy (CRediT) which was developed in 2014. CRediT tool has an advantage over existing methods by being more detailed and structured. CRediT defines 14 different types of author contributions. If this method would be widely adopted by publishers, it could help to provide a better documentation of the contribution of researchers. Finally, the paper presents the new web app and R package (tenzing) which helps to facilitate CRediT tool.

The authors aimed:

1. To revise the current state of how journals use CRediT tool for documentation of author contribution and difficulties associated with the use of CRediT during submission process.

2. To present R package and web app (tenzing) which they developed to overcome these difficulties.

In my opinion, the authors have achieved both aims. The paper summarizes how some journals are currently using new author contribution documentation tool CRediT, and explains difficulties of using it based on illustrative examples. Free web app and R package tenzing, developed by authors, achieves its purpose. Figures included in the paper are very informative and help to understand how tenzing package works.

I find the paper very interesting and I think it would be beneficial to replace traditional author contribution methods by more sophisticated methods such as CRediT in the future.

If the structure of the paper meets the requirements of PLUS ONE journal, then I would suggest accepting the paper.

Reviewer #2: Dear authors,

I thoroughly enjoyed reading your manuscript titled "Documenting contributions to scholarly articles using CRediT and tenzing", discussing the CRediT standard and the associated tenzing R package. The issue of authorship or “contributorship” is timely and deserving of more attention, and the development of software packages and web applications is – in my view – an excellent way of raising discussion while also providing a solution. I only have a few minor suggestions that I hope could improve the manuscript, and the app, even further.

Section: The future of contributorship

It would be interesting to see a bit more discussion raised concerning large(r)-scale projects and group authorship vis-à-vis the CRediT guidelines (e.g., Fontanarosa, Bauchner, & Flanagin, 2017; https://jamanetwork.com/journals/jama/fullarticle/2667044). Given that assigning the correct role to authors and collaborators has proven to be difficult, is crediting the entire research group as a whole (and thus no individual authors) perhaps a viable alternative? The specific roles (following, e.g., CRediT guidelines) of each author could then be specified on the group’s website, for instance.

Section: The future of CRediT

It is unfortunate that the “lead”, “equal”, and “supporting” specifications are not more widely adopted, and thus not a feature of tenzing. While I understand your reasons, I hope that you may add such a feature in coming releases. Perhaps showing how it can be done in practice will lead to faster adoption?

The tenzing app

I tested the Shiny app (using Firefox) with an edited template file (edited with LibreOffice Calc on Pop_OS! 20.04 Linux, saved as xlsx). All features worked as intended, with one exception. The papaja YAML output fails to retrieve all roles; all roles are stated as "Conceptualization" (see attachments). I had a look at the code and could not find any obvious source for this error (overall, while browsing through the GitHub repo, I found the code to be clean, consistent, and nicely commented).

While the app works well, apart from the above error, I have two feature suggestions that may encourage quicker adoption by users:

(1) Allow users to create a template from scratch within the Shiny app (thus not having to download a template, edit it, and then upload it). It's a minor issue, but it is always nice to avoid having to move from one program to another. This could be done on a separate tab or page, and the final sheet could shown as a datatable, or similar, to allow editing within the browser (similar to the “Show infosheet” feature, but with editing capabilities). A download button would allow users to save the sheet locally, if need be.

(2) It would be helpful to allow for more than two affiliations. Perhaps this could be a dynamic variable?

Formatting errors

• Line 49, missing abbreviation (ISWC) following mention of International Standard Musical Work Code.

• Line 70, CRediT is misspelled as CrediT.

• Line 169, which is misspelled as whichh.

Best of luck,

Carl Delfin

Reviewer #3: The authors describe a very important tool that appears to solve an emerging problem about authorship and contributions to scholarly articles. Their recommended tool is useful, and would help to initiate change in the way in which scientist conduct and report research/outputs.

However, the “CRediT and tenzing” tool in itself does not solve all the problems. For instance, one challenge that has not been well elucidated is how to fairly document the depth of involvement by an author per category. This will help resolve issues where some authors may contribute to a very great depth in just one category, which may surpass the total contributions from other authors who would be checking the boxes in more categories on the CRediT scale.

Reviewer #4: Comments to the authors:

In this report, the authors discussed the issue with authorship position in scientific publications, which can carry little information about the actual contribution of each co-author. Authors briefly review the emerging CRediT standard for documenting contributions and discuss their web app and R package “tenzing” application that facilitates researcher reporting of contributorship information in manuscripts and journal articles. The topic itself is interesting and scientifically relevant, and could be suitable for the scope of the journal. Some minor comments for the authors:

- The manuscript still needs editing and language revision.

- The manuscript lacks some important references, for example, in the “introduction” section and “how journals are using CRediT”.

- The facility of PLoS journals can be provided as a separate figure e.g. Figure 2 (as for eLife)

- Instead of links to external blogs, I recommend authors to provide tables describing different features of CRediT e.g. limitations and applicability.

Reviewer #5: Thank you for allowing me to read this article in which the authors describe the challenges by informing who did what in the project manuscript. Lack of standardization on how to report contributorship in multi-authored published scholarly works is one of the issues. The authors find this to be particularly problematic since researchers indicate contributions late in the process, namely when submitting a manuscript. They suggest that developing a structured format would help, for instance, be more transparent, and give researchers specific recognition.

Authors describe the use of Contributor Roles Taxonomy (CRediT), developed in 2014, and widely utilized by scientific journals. Authors created a web-based app tenzing designed to make it easier for researchers to plan and record contributions of each team member. The presentation of tenzing is relatively short. Tenzing is a pre-filled Google Sheet (or Excel) template containing all 14 CRediT categories and rows for entering the collaborator’s information. The authors present tenzing as a component that can be used with an existing system (via the underlying R package). Uploading the spreadsheet to the application enables to generate various outputs. The authors provide pictures where they illustrate the use of tenzing on their project. There is no information if other research groups have tested the application.

I would appreciate having more information on how tenzing is advantageous? In what way is this method more superior to existing ones? Is there any possibility to trace the changes researches make in tenzing?

Many different components of the same problem emerge in most parts of the manuscript, which illustrates how multidimensional the question is. At the same time, it would be clearer if authors were to focus on a limited inquiry, what part of the problem do authors want to solve? Another possible alternative is to choose a different article type.

I find the structure of this article unclear. The authors should clarify the following sections: problem definition, proposed solution, discussion, conclusion.

The main strength of this manuscript is that it addresses a timely question with great potential for improvement. I agree with the authors that there is a need to develop methods and strategies to ensure transparency in the research were contribution plays a significant part. But still, there is an unresolved question regarding what approach would solve the particularly chosen part of the issue.

6. PLOS authors have the option to publish the peer review history of their article (what does this mean?). If published, this will include your full peer review and any attached files.

Reviewer #1: No

Reviewer #2: **Yes: **Carl Delfin

Reviewer #3: No

Reviewer #4: No

Reviewer #5: No

---

## [Author Response · Author response to Decision Letter 0]

15 Oct 2020

Dear Dr. Nwaru,

We greatly appreciate the five reviews for our manuscript. This provides a lot of value, and not only due to the constructive suggestions - thanks to PLoS’s option for open peer reviews, which we plan to take up, it also provides a public indication of the extent to which our manuscript was vetted and evaluated.

Response to comment by Napsi Szincsak of PLoS ONE

After the action letter, I received an email from Napsi Szincsak asking us to address the fact that PLoS ONE has specific guidelines on software sharing. We had already cited the Github page, where the source code is archived. We have now also added an MIT license (https://github.com/marton-balazs-kovacs/tenzing/blob/master/LICENSE.md).

Response to “journal requirements” comments

-1. Please ensure that your manuscript meets PLOS ONE's style requirements, including those for file naming.

We have adjusted the title page and believe the rest of the manuscript and files comport with the specified style.

-2. In your Data Availability statement, you have not specified where the minimal data set underlying the results described in your manuscript can be found. 

That is correct – this is because there is no underlying data set, as there are no results described in the manuscript.

Response to reviewer 1’s comments

We appreciate that the reviewer believes the manuscript is interesting and that it achieves its aims of reviewing the state of author/contribution issues and presenting the R package and app for improving things.

Response to reviewer 2’s comments

We appreciate the reviewer’s praise for the manuscript as well as the multiple detailed “minor suggestions”.

It would be interesting to see a bit more discussion raised concerning large(r)-scale projects and group authorship vis-à-vis the CRediT guidelines (e.g., Fontanarosa, Bauchner, & Flanagin, 2017; https://jamanetwork.com/journals/jama/fullarticle/2667044). Given that assigning the correct role to authors and collaborators has proven to be difficult, is crediting the entire research group as a whole (and thus no individual authors) perhaps a viable alternative? The specific roles (following, e.g., CRediT guidelines) of each author could then be specified on the group’s website, for instance.

We agree that group (or consortium, as it is sometimes called) authors is a viable alternative to contributor schemes such as CRediT and use of them has become a strong tradition in some fields, such as genomics. Documenting CRediT information on the group’s website, however, would not be considered a great solution as websites are typically not as permanent as journal article records. We were not aware of the JAMA editorial the reviewer provided, which is helpful, so we have added a paragraph on this with that citation to this section.

It is unfortunate that the “lead”, “equal”, and “supporting” specifications are not more widely adopted, and thus not a feature of tenzing. While I understand your reasons, I hope that you may add such a feature in coming releases. Perhaps showing how it can be done in practice will lead to faster adoption? 

Indeed, we do think that this is very important to specify in some projects, see some of our discussion here https://github.com/marton-balazs-kovacs/tenzing/issues/15 . As mentioned at the discussion, there is presently no JATS (the underlying machine-readable metadata) specification in CRediT for lead vs. equal vs. supporting, but we expect that the group at the National Information Standards Organization (NISO) presently working on CRediT will address this, which will lead us to move this feature up in the development plan.

I tested the Shiny app… The papaja YAML output fails to retrieve all roles; all roles are stated as "Conceptualization".

This error has since been fixed. 

I found the code to be clean, consistent, and nicely commented

We appreciate your gracious comment.

I have two feature suggestions that may encourage quicker adoption by users:

(1) Allow users to create a template from scratch within the Shiny app (thus not having to download a template, edit it, and then upload it). It's a minor issue, but it is always nice to avoid having to move from one program to another. This could be done on a separate tab or page, and the final sheet could shown as a datatable, or similar, to allow editing within the browser (similar to the “Show infosheet” feature, but with editing capabilities). A download button would allow users to save the sheet locally, if need be.

We have been discussing this feature, but implementing spreadsheet functionality within Shiny is currently beyond our programming resources as well as, possibly, the Shiny hosting infrastructure we have access to. However, we have been exploring the feasibility and usability of users entering into Tenzing the Google Doc URL for their contributorship spreadsheet so that users do not need to download and upload a template.

(2) It would be helpful to allow for more than two affiliations. Perhaps this could be a dynamic variable?

Yes, we have been working on this feature for a while, actually, trying different formats for the interface to avoid creating major usability problems. We hope to have it implemented in the next few months.

• Line 49, missing abbreviation (ISWC) following mention of International Standard Musical Work Code

•Line 70, CRediT is misspelled as CrediT.

• Line 169, which is misspelled as whichh.

Thanks for pointing these out – we have fixed them.

Response to Reviewer 3’s comments

The authors describe a very important tool that appears to solve an emerging problem about authorship and contributions to scholarly articles. Their recommended tool is useful, and would help to initiate change in the way in which scientist conduct and report research/outputs.

Thank you for the positive comments about tenzing.

one challenge that has not been well elucidated is how to fairly document the depth of involvement by an author per category.

We agree that this remains a big challenge. The CRediT standard itself has a way to partially address it, by allowing indication of whether a contributor made a “leading”, “equal”, or “supporting” role, but it is not yet fully implemented by CRediT: There is presently no JATS (the underlying machine-readable metadata) specification in CRediT for lead vs. equal vs. supporting, but we expect that the group at the National Information Standards Organization (NISO) presently working on CRediT will address this, which will likely prompt us to move this feature up in the development plan (see some of our discussion here https://github.com/marton-balazs-kovacs/tenzing/issues/15).

Response to Reviewer 4’s comments

The topic itself is interesting and scientifically relevant, and could be suitable for the scope of the journal. Some minor comments for the authors:

Thank you for the positive comments.

- The manuscript still needs editing and language revision.

We have made many edits to improve the wording, as can be seen in the track changes version.

- The manuscript lacks some important references, for example, in the “introduction” section and “how journals are using CRediT”.

Thank you for pointing this out - we have now added several references to these sections.

- The facility of PLoS journals can be provided as a separate figure e.g. Figure 2 (as for eLife)

Done.

- Instead of links to external blogs, I recommend authors to provide tables describing different features of CRediT e.g. limitations and applicability.

Thank you for the suggestion. We have added a table describing the basic features of CRediT (Table 1), which also provides some indication of CRediT’s limitations.

Response to Reviewer 5’s comments

I would appreciate having more information on how tenzing is advantageous? In what way is this method more superior to existing ones?

One reason that this aspect of the manuscript may have appeared lacking is because we don’t know of any alternative methods to tenzing, in the sense of an easy to use (no programming) tool to facilitate CRediT reporting. To address the reviewer’s comment, however, we have added more references to somewhat related approaches, and have added a paragraph about “authorship grids” to better explicate the unique utility of tenzing.

Is there any possibility to trace the changes researches make in tenzing?

The researchers enter the contributorship information into a Google Sheet (or they can download that and make changes in a spreadsheet of their choice). Google provides a version history that can be reviewed, for example to understand how contributorship roles evolve during the course of the project.

Many different components of the same problem emerge in most parts of the manuscript, which illustrates how multidimensional the question is. At the same time, it would be clearer if authors were to focus on a limited inquiry, what part of the problem do authors want to solve? Another possible alternative is to choose a different article type.

I find the structure of this article unclear. The authors should clarify the following sections: problem definition, proposed solution, discussion, conclusion.

The main strength of this manuscript is that it addresses a timely question with great potential for improvement. I agree with the authors that there is a need to develop methods and strategies to ensure transparency in the research were contribution plays a significant part. But still, there is an unresolved question regarding what approach would solve the particularly chosen part of the issue.

We tentatively agree that doing nearly a complete rewrite of the manuscript with the structure proposed by the reviewer should make it clearer to some readers. However, given that the other four reviewers did not suggest that a rewrite was necessary, we prefer not to make too many changes that might perturb the manuscript in a way that might make the other reviewers less happy. We also agree that ideally, an argument could be made that tenzing is the optimal way to address the needs of researchers and the interests of science broadly. However, we make no such claim in the manuscript – instead we aimed only to explain some of the issues and that tenzing partially addresses some of them. We think that the limited rewrite that we have done makes that a bit more clear.

---

## [Decision Letter · Decision Letter 1]

14 Dec 2020

Documenting contributions to scholarly articles using CRediT and tenzing

PONE-D-20-23271R1

Dear Dr. Holcombe,

We’re pleased to inform you that your manuscript has been judged scientifically suitable for publication and will be formally accepted for publication once it meets all outstanding technical requirements.

Kind regards,

Cassidy R Sugimoto, Ph.D.

Academic Editor

PLOS ONE

Additional Editor Comments (optional):

Reviewers' comments:

Reviewer's Responses to Questions

**Comments to the Author**

1. If the authors have adequately addressed your comments raised in a previous round of review and you feel that this manuscript is now acceptable for publication, you may indicate that here to bypass the “Comments to the Author” section, enter your conflict of interest statement in the “Confidential to Editor” section, and submit your "Accept" recommendation.

Reviewer #2: All comments have been addressed

Reviewer #3: All comments have been addressed

Reviewer #4: All comments have been addressed

2. Is the manuscript technically sound, and do the data support the conclusions?

Reviewer #2: Yes

Reviewer #3: Yes

Reviewer #4: (No Response)

3. Has the statistical analysis been performed appropriately and rigorously? 

Reviewer #2: N/A

Reviewer #3: N/A

Reviewer #4: (No Response)

4. Have the authors made all data underlying the findings in their manuscript fully available?

Reviewer #2: Yes

Reviewer #3: No

Reviewer #4: (No Response)

5. Is the manuscript presented in an intelligible fashion and written in standard English?

Reviewer #2: Yes

Reviewer #3: Yes

Reviewer #4: (No Response)

6. Review Comments to the Author

Reviewer #2: Dear authors,

Thank you for addressing each of my concerns and suggestions. I believe that this is an important and valuable study in the move towards more transparent and open research practices. Best of luck in the future!

Kind regards,

Carl Delfin

Reviewer #3: The authors have attempted to address all the issues raised, and justifiably, have done so satisfactorily.

Reviewer #4: (No Response)

7. PLOS authors have the option to publish the peer review history of their article (what does this mean?). If published, this will include your full peer review and any attached files.

Reviewer #2: **Yes: **Carl Delfin

Reviewer #3: No

Reviewer #4: No

---

## [Editor Report · Acceptance letter]

21 Dec 2020

PONE-D-20-23271R1 

Documenting contributions to scholarly articles using CRediT and tenzing 

Dear Dr. Holcombe:

I'm pleased to inform you that your manuscript has been deemed suitable for publication in PLOS ONE. Congratulations! Your manuscript is now with our production department. 

Kind regards, 

on behalf of

Dr. Cassidy R Sugimoto 

Academic Editor

PLOS ONE